

# Direct, indirect, and vicarious nature experiences collectively predict preadolescents' self-reported nature connectedness and conservation behaviors

Zhihui Yue[1,2] and Jin Chen[1]

[1] CAS Key Laboratory of Tropical Forest Ecology, Xishuangbanna Tropical Botanical Garden, Chinese Academy of Sciences, Mengla, Yunnan, China
[2] University of Chinese Academy of Sciences, Beijing, China

## ABSTRACT

**Background:** The quantity and context of children's nature experiences are undergoing significant changes, exacerbating a pervasive negative cycle that could impact future conservation efforts. Therefore, it is essential to conduct further studies on the potential impacts of these changes on children's willingness to engage in conservation practices.

**Methods:** We surveyed 2,175 preadolescents (aged 9–12) from rural and city schools in Hangzhou, Kunming, and Xishuangbanna, China, regarding their nature experiences (direct, indirect, and vicarious) and self-reported nature connectedness and conservation behaviors.

**Results:** We found that children in urban areas have higher frequencies of indirect and vicarious experiences than those in rural areas, with some direct nature experiences seldom reported among city respondents. Direct, indirect, and vicarious nature experiences significantly predicted children's conservation behavior and collectively provided the highest predictive power for conservation behavior. Direct and vicarious experiences were strongly correlated with pro-nature behavior, and the latter with pro-environmental behavior. Emotional and cognitive connection with nature positively predicted conservation behavior, influenced by location and residence type.

**Discussion:** This study reveals that different types of nature experiences shape children's current conservation behaviors in China.

## INTRODUCTION

The rate of global habitat degradation and species extinction are at unprecedented levels in human history (*Intergovernmental Science-Policy Platform on Biodiversity and Ecosystem Services (IPBES), 2018*), and the global scientific community has reached a consensus that the record acceleration of these ecological crises is human-driven (*Ceballos et al., 2015*). To aid nature's recovery, there is an urgent need to understand and promote pro-nature conservation behavior. In the past decades, two main strategies for biodiversity

Corresponding author
Jin Chen, cj@xtbg.org.cn

conservation have been adopted: (1) separating people from nature to protect it; and (2) reconnecting people with nature (*Sandbrook, 2015*). The latter, in which the framing and purpose of conservation takes the view of "people and nature," emphasizes a nuanced and dynamic relationship between the two forces and has become a new trend in recent years (*Mace, 2014*). This view focuses on a shared environment where the form, function, adaptability, and resilience provided by nature are highly valued (*Chan et al., 2016*). Therefore, ensuring the connection between people and nature is considered critical for an effective conservation strategy (*Butchart et al., 2010*).

However, a common challenge in maintaining the "people and nature connection" is that modern-day children are losing interest in and experience with the natural world (*Soga & Gaston, 2018*). This disconnect between children and nature is a cause for concern from both natural and environmental perspectives (*Bragg et al., 2013*). The decline in the experience to nature, particularly to biodiversity (*Beery & Jørgensen, 2018*), can lead to decreased cognitive and affective connection with nature, exacerbating an already pervasive negative cycle, and potentially impeding future conservation efforts (*Soga et al., 2016*; *Bashan, Colléony & Shwartz, 2021*). Since nature experiences undergo culturally driven transformations (*Clayton et al., 2017*; *Novotný et al., 2021*), the experiences of earlier generations may not necessarily be replicated in later generations; therefore, it is critical to reconsider the definition and framing of nature experiences in modern societies.

## Various nature experiences and ongoing changes

Experiences are situations in which a person is engaged in an interaction at an emotional, physical, spiritual, or intellectual level in terms of knowledge, skills, attitudes, and behavior (*Clayton et al., 2017*). Nature experiences include individuals' perceptions and/or interactions with stimuli from the natural world through a variety of sensory modalities (*Hartig et al., 2011*; *Bratman et al., 2019*). Notably, nature experience is described not only by its real content but also by its immediate social context, which is shaped by the surrounding society and culture (*Clayton et al., 2017*). Global urbanization (*Lin et al., 2014*) and the development of technology (*Korfiatis, Photiou & Petrou, 2020*), coupled with new working methods and leisure activities (*Hartig & Kahn, 2016*; *Edwards & Larson, 2020*), significantly affect children's views and experiences of nature (*Mcphie & Clarke, 2020*). A benign by-product of modernity is that most individuals do not depend directly on harvesting natural resources for their survival; instead, they have developed new types of nature experiences while redefining their place in nature (*Pilgrim et al., 2008*). Unlike past generations, who spent considerable time in the natural world to sustain their livelihoods, modern children interact with the natural world because they want to (*Oh et al., 2020*; *Novotný et al., 2021*). Therefore, the purpose and forms of children's interactions with nature have shifted to multiple, diverse, and dispersed formats (*Edwards & Larson, 2020*).

Nature takes on many forms and there are many ways to interact with it. *Keniger et al. (2013)* listed six settings (indoor, urban, fringe, production landscape, wilderness, and specific species) with three types of nature experiences (indirect, incidental, and intentional). To further segment these, *Clayton et al. (2017)* proposed six dimensions of

nature experiences: observing *vs* interacting, consumptive *vs* appreciative, self-directed *vs* other-directed, separate *vs* integrated, solitary *vs* shared, and positive *vs* negative. Nature experiences can also be divided more broadly into direct, indirect, and vicarious experiences (*Kahn & Kellert, 2002*). Experiences in nature can be direct and intentional (*i.e.*, direct nature experiences), involving actual contact with natural settings through exploration, play, and activities in green areas where people experience wildlife without human control (*Freeman & Tranter, 2012*; *Rosa & Collado, 2019*). By contrast, indirect nature experiences refer to those involving physical contact with nature in managed, contrived, or restricted contexts such as arboretums, botanical gardens, aquariums, zoos, museums, and nature centers (*Kahn & Kellert, 2002*). Finally, vicarious experiences refer to a wide range of activities that occur in the absence of actual physical contact with nature but with various representations of nature, including realistic, metaphorical, symbolic, or stylized characterizations viewed *via* TV, movies, computers, magazines, or books (*Kahn & Kellert, 2002*). Given that different types of nature experiences may produce distinctive impacts that shape children's nature cognition, their roles in promoting human-nature connection and future conservation behavior require further investigation (*Truong & Clayton, 2020*).

Numerous studies have shown that most children today spend more time on electronic media than on outdoor activities (*Larson et al., 2019*; *Edwards & Larson, 2020*). While emphasizing the role of mediated experiences, *Novotný et al. (2021)* point out those contemporary children have more overall experience with nature than those of the early 20th century. Hence, children's nature experiences can occur through conditions of real contact, representations, or simulations. They may be deliberate or incidental, affected by personal connections and sociocultural meanings, and may occur in the context of diverse activities (*Bratman et al., 2019*). The myriad of nature experiences children experience today certainly provides a unique and diversified space that shapes their nature cognition (*Mcphie & Clarke, 2020*), thus supporting the call for understanding the effect such experiences have on children's conservation willingness and behaviors.

## Factors that influence conservation behaviors

Understanding the individual and contextual factors that influence conservation behaviors is vital to encourage and boost engagement in such activities (*Richardson et al., 2020*). However, the focus of conservation behavior often varies under different circumstances, and behaviors addressing biodiversity issues need to be specified (*Prévot et al., 2018*). To date, behaviors related to directly supporting habitats for wildlife, that is, pro-nature behaviors, are often ignored in most pro-environmental behavior scales (*Gkargkavouzi, Halkos & Matsiori, 2019*). Yet, pro-nature behaviors are distinct from pro-environmental behaviors (*Richardson et al., 2020*). Chiefly, pro-environmental behaviors typically focus on resource use and energy saving, whereas pro-nature behaviors center on wildlife-oriented actions (*Hughes, Richardson & Lumber, 2018*). Thus, effective engagement with conservation needs to encourage behaviors geared toward not only reducing our impact on the earth but also directly supporting the restoration of biodiversity and habitats (*Richardson et al., 2020*; *Austen et al., 2021*).

Constructivism learning theory recognizes child development and knowledge acquisition as processes that are shaped by socio-cultural experiences (*Cunningham, 1991*). Children's understanding of nature and their relationship with nature is based on the interplay between past and present experiences and is ever changing and developing (*Adams & Savahl, 2017*). Previous studies have indicated that engaging with and experiencing the complexity and variation of nature through self-directed and rich body-sensory immersive play is key to gaining a deeper understanding of biodiversity (*James & Bixler, 2008*; *Beery & Jørgensen, 2018*; *Abdullah, Ishak & Ahmad, 2022*); indeed, it may stimulate greater engagement in conservation-friendly behaviors (*Beery & Jørgensen, 2018*; *Richardson et al., 2020*). Direct interactions with diverse living and non-living aspects of nature can enhance people's comprehension of biodiversity and increase their personal relevance (*Bixler, Floyd & Hammitt, 2002*). Furthermore, direct nature experiences are likely to be more vivid and multisensory, thus eliciting strong emotions and creating a lasting and positive memory (*Cooper et al., 2015*; *Asah et al., 2018*), which may, in turn, enhance children's conservation willingness *via* the mediation of biophilia (*Zhang, Goodale & Chen, 2014*). Children's "hands on" nature experiences are crucial for the development of specific skills, such as of observing and classifying organisms (*Klofutar, Jerman & Torkar, 2020*).

Indirect nature experiences are probably the most common means of interacting with nature in cities and may lead to positive conservation behaviors (*Hughes, 2013*; *Oh et al., 2021*). Zoological facilities, such as zoos, aquariums, zoological theme parks, and wildlife centers offer more accessible interaction opportunities, and the endangered, threatened, or rare animals they support can affect biodiversity knowledge and motivate behavioral change (*Ogden & Heimlich, 2009*; *Mann, Ballantyne & Packer, 2018*; *Huang et al., 2019*; *Collins et al., 2020*). Botanical gardens consisting of natural or semi-natural remnant land patches immersed in large urban areas are important to enhance people's knowledge of plants, improve awareness of biodiversity, and support research and conservation efforts (*Williams et al., 2015*; *Suárez-López & Eugenio, 2018*; *Zelenika et al., 2018*). However, some studies have reported that while indirect nature experiences may have a greater impact on knowledge than direct ones, such effects are marred by better knowledge of exotic species rather than of the locally abundant wildlife (*Genovart et al., 2013*; *Almeida, Fernández & Strecht-Ribeiro, 2018*). Therefore, the effect of indirect nature experiences on conservation behavior requires further investigation.

Vicarious experiences provide enjoyment and benefits that stimulate people's interest in seeking real-world nature experiences and supporting conservation (*Truong & Clayton, 2020*). Engaging in more passive activities of watching or listening to nature designed to capture unpredictable or less visible wildlife living in inaccessible terrains or extreme weather promotes awareness of particular species or encourage public support (*Verma, van der Wal & Fischer, 2015*; *Louson, 2021*). Studies suggest that mediated experiences might positively impact children's affective attitudes and willingness to conserve biodiversity (*Soga et al., 2016*), but they are likewise often more prone to protecting unseen and exotic rather than local animal species (*Ballouard, Brischoux & Bonnet, 2011*). In addition, content focusing narrowly on familiar species and "spectacular" narratives

(*Somerville et al., 2021*) may result in stereotypical representations of certain species (*Crowley, Silk & Crowley, 2021*) or produce romantic constructions of nature as pristine wildness (*Mcphie & Clarke, 2020*), which hinders the success of communication and conservation goals.

Connection with nature is, in the broader sense, the idea that humans see themselves as part of nature, which can be used as leverage to achieve conservation (*Barragan-Jason et al., 2022*). Indeed, connection with nature plays a central role, in conjunction with other factors, in promoting pro-environmental (*Otto & Pensini, 2017*; *Bruni et al., 2018*; *Mackay & Schmitt, 2019*) and pro-nature (*Prévot et al., 2018*; *Richardson et al., 2020*) behaviors. Furthermore, the outcomes of various nature experiences are partly mediated by the extent of individuals' connection with nature (*Mayer et al., 2009*; *Skibins, Powell & Hallo, 2013*; *Arendt & Matthes, 2016*). Children and adolescents exhibiting higher levels of nature connectedness demonstrate greater environmental knowledge (*Cheng & Monroe, 2012*; *Otto & Pensini, 2017*) and report more pro-nature behaviors, such as putting out food for birds and joining a nature club (*Hughes, Richardson & Lumber, 2018*; *Richardson et al., 2020*). They also have more routine pro-environmental behaviors (*Barrera-Hernández et al., 2020*).

Socio-demographics such as age, gender, and geographic locations are important variables for predicting children's nature experiences and engagement in conservation behavior. Studies indicate that nature experiences during childhood, particularly before the age of 12, can lead to a stronger relationship with nature in adulthood (*Liefländer et al., 2013*; *Rosa, Profice & Collado, 2018*). Positive engagement with nature during childhood is also a predictor of pro-environmental behavior in adulthood (*Evans, Otto & Kaiser, 2018*; *Molinario et al., 2020*). Moreover, environmental attitudes and behaviors begin to develop in early childhood and continue through early adolescence, peaking at around age 10 and stabilizing by age 14. Preadolescents between the ages of 9 and 12 are at a critical developmental stage, where they can establish a positive relationship with nature and develop cognitive abilities necessary for understanding ecological systems (*Piaget et al., 1969*; *Kahn & Kellert, 2002*). Gender also has a significant effect on school-age children's conservation attitudes and willingness (*Soga et al., 2016*). Efforts to protect nature also need to consider geographical locations. The gradient of local biodiversity and urbanization may affect children's nature experiences (*Hanisch, Johnston & Longnecker, 2019*; *Bashan, Colléony & Shwartz, 2021*) and conservation values (*MacDonald et al., 2015*).

Modern societies offer many means to modify, replace, complement, or expand children's nature experiences in various ways (*Soga et al., 2016*; *Jeon, Yeon & Shin, 2018*; *Klofutar, Jerman & Torkar, 2020*). However, not all outcomes of nature experiences are equivalent, and perhaps not all have even been described (*Truong & Clayton, 2020*). Overall, research on changes in the types and content of modern children's nature experiences that simultaneously acknowledge the challenges in exploring the relationship between multiple, diverse, and dispersed nature experiences and conservation behavior remains largely lacking. Moreover, evidence examining the path from children's nature

experiences to their connection with nature and conservation behaviors is still thin; hence, a more nuanced understanding of the chain relationship between them is needed.

China is a country with immense biodiversity that faces severe threats (*Zhang et al., 2022*). Along with rapid urbanization and lifestyle change, the Chinese government has put forward the concept of "ecological civilization," aimed at striking a balance between economic benefits and ecosystem sustainability (*Dong et al., 2021*; *Li et al., 2022b*) while seeking overall well-being for both people and the planet. To this end, thousands of nature-based outdoor education programs aimed at connecting children with nature and promoting biodiversity conservation have been initiated by different organizations across China. Additionally, many botanical gardens and zoological facilities are prioritizing educating visitors, particularly school-age children (*Roe & McConney, 2015*). As in many other regions in the world, Chinese children experience a modified natural environment and nature experiences with mixed forms.

## Study purpose

Understanding the nature experiences of the new generation and reconnecting children with nature are essential in current and future biodiversity conservation. In this study, taking into account the critical period for children to develop environmental relationships and behaviors, we investigated the components of preadolescents' (aged 9–12) nature experiences in China that aid in constructing children's connections and behaviors toward nature. Our study aimed to address the following research questions: (1) What are the nature experiences of contemporary Chinese preadolescents? (2) How do different types of nature experiences support children's connection with nature and conservation behavior? (3) How do urbanization and geographic location affect children's nature experiences, connection with nature, and conservation behavior? Building upon prior research and literature review, we formulated the hypothesis that (1) Chinese prepubescent children's experiences with nature are diverse, including direct, indirect, and vicarious encounters; (2) direct, indirect and vicarious nature experiences can positively predict children's nature connectedness and conservation behavior; and (3) the type of residence and geographic location may act as moderators in the association between nature experience, nature connectedness, and conservation behavior. For instance, students residing in rural areas are more likely to have direct encounters with nature, leading to higher levels of nature connectedness and conservation behavior. Likewise, students from regions abundant in biodiversity may exhibit greater conservation behavior.

## MATERIALS AND METHODS

We conducted our study in two phases from June 2020 to April 2021. First, to explore Chinese preadolescents' nature experience, we opportunistically sampled four primary schools (two rural and two urban primary schools) in Xishuangbanna to conduct focus group interviews. Two pilot tests were performed to revise and refine the research questionnaires. Second, considering the variations in urbanization and biodiversity (Table S1), we selected three locations (Hangzhou, Kunming, and Xishuangbanna) to conduct a questionnaire survey. According to administrative divisions in China, our city

**Table 1 Demographics of participants for questionnaire (N = 2,175).**

| Variables | Count | Percentage (%) |
|---|---|---|
| *Gender* | | |
| Boy | 1,095 | 50.3 |
| Girl | 1,080 | 49.7 |
| *Age* | | |
| 9-year-old | 399 | 18.3 |
| 10-year-old | 590 | 27.1 |
| 11-year-old | 615 | 28.3 |
| 12-year-old | 571 | 26.3 |
| *Residence* | | |
| Rural | 1,032 | 47. 5 |
| City | 1,143 | 52.6 |
| *Location* | | |
| Xishuangbanna | 731 | 33.6 |
| Hangzhou | 584 | 26.9 |
| Kunming | 860 | 39.5 |

samples are from the administrative districts above the county level, while rural areas refer to administrative districts including towns, townships, and villages. Based on random sampling, 15 public elementary schools from urban and rural areas were included, and preadolescents aged 9–12 in grades 3–5 participated in the study. A total of 2,295 questionnaires were collected, of which 2,175 valid questionnaires were retained (Table 1).

## Study area

Three locations were involved in our study: Xishuangbanna, Kunming, and Hangzhou. Xishuangbanna, located in the Mekong Region, southwest Yunnan Province, southwest China, lies in the Indo-Burma global biodiversity hotspot, representing an area with high biodiversity and relatively less economic development. Xishuangbanna contains a remarkable diversity of species—while covering only 0.2% of the land area of China. It harbors 16% of the vascular flora, 21.7% of the mammals, and 36.2% of the birds found in the country. Kunming, which has intermediate urbanization and economic development, is the capital of southwest China's Yunnan province, where the *15th Conference of the Parties to the Convention on Biological Diversity* was held in 2021. For comparison, we also chose Hangzhou to present a developed region in China. Hangzhou, an important city located in eastern China and on the Yangtze River Economic Belt, had an urbanization rate of 83.6% by the end of 2021.

## Focus group interviews

Focus groups present a more permissive and conversational setting in which children may communicate without literacy barriers to facilitate speaking about their understanding and experiences (*Barker & Weller, 2003*). In this study, we conducted focus group interviews

with 12 (six boys and six girls) third-grade students from Menglun Central Primary School, selected through convenience sampling, to assess children's understanding of these questions. Based on the comprehensibility of these questions, we further selected 12 children (six boys and six girls) from grades 3 to 5 in each of the four schools for interviews. Each focus group lasted 30–40 min and was facilitated by the same researcher. The semi-structured focus group elicited children's views of nature experiences through a series of questions such as "Where can you find nature?"; "What activities are usually done in nature?" and "How can you learn about nature besides going to nature?" The researcher asked questions, and the children took turns answering and discussing. Thirteen focus group discussions ($N = 156$) were held with the participants, all of which were audio recorded, transcribed *verbatim*, and double-checked for accuracy to ensure trustworthy data.

## Questionnaire

### Pilot test

Two pilot tests were conducted in Menglun ($N = 277$) and Jinghong ($N = 268$) to develop and refine the scales and data collection procedures. The scales were translated following the four-step method of scale translation and reverse translation (*Harkness & Schoua-Glusberg, 1998*). The translated scales underwent initial testing among preadolescents aged 9–12 years, while participating children were randomly interviewed. We removed three items from the Children's Connection to Nature Index that aimed to express a sense of responsibility (*e.g.*, "Picking up trash on the ground can help the environment") due to a strong ceiling effect. Items with lower factor loadings (<0.3) in Nature experience Scale ("Travel to natural scenic spots"; "Take walk outdoors" and "Picnicking") were deleted. Factor analysis largely confirmed the three theoretical constructs with the modifications of our Nature Experience Scale (Kaiser-Meyer-Olkin = 0.914, explaining 53.15% of the variance). Reliability among scale items was measured using Cronbach's $\alpha$. The reliability of the three subscales (Nature experience Scale with $\alpha = 0.889$; Children's Connection to Nature Index with $\alpha = 0.717$; The Children's Biodiversity Conservation Scale with $\alpha = 0.756$; $N = 545$) was all acceptable. Based on the reliability and validity of the scales, we used the revised translation scale in the follow-up survey.

### Nature experience

We extracted children's frequently mentioned nature experience items from the focus group interviews. After combining the Chinese Children's Natural Experience Scale (*Zhang, Goodale & Chen, 2014*) and the Children's Different Forms of Natural Experience Scale (*Mustapa et al., 2018*), 20 items were developed to measure self-reported nature experience, including direct (11 items), indirect (five items), and vicarious (four items) nature experience. The scale was tested in pilot tests, and 17 items were reserved for formal investigation. Participants were instructed to indicate the frequency of participation in different forms of nature experience in the past year using a 5-point Likert scale (never = 1, rarely = 2, sometimes = 3, often = 4, always = 5).

### Connection with nature

Despite differences in the definitions of connection with nature, there is a common underlying construct (*Tam, 2013*). Connection with nature metrics that include both cognitive and affective aspects showed significantly higher effect sizes than metrics with only cognitive components (*Barragan-Jason et al., 2022*). Therefore, the 16-item Children's Connection to Nature Index (CNI) (*Cheng & Monroe, 2012*) was used to measure self-reported connections with nature. The original scale comprising 16 questions divided into four sections was revised and 13 items were retained for two dimensions (cognitive and emotional) to apply to Chinese children. Respondents were asked to evaluate the degree of approval of the 13 items by ticking one of the following declarations for each: "Strongly disagree," "Disagree," "Not sure," "Agree," or "Strongly agree".

### Conservation behavior

The Children's Biodiversity Conservation Scale (*Hughes, Richardson & Lumber, 2018*) consisting of six items on pro-environmental (*Collado et al., 2015*) and seven items on pro-nature children's behavior was adopted, and a 5-point Likert scale (never = 1, rarely = 2, sometimes = 3, often = 4, always = 5) was used to rate the preadolescents' participation frequency in those behaviors in the past year.

### Questionnaire procedure in 15 schools

Questionnaire surveys were conducted on school days with permission from each school. We randomly selected one class among grades 3–5 in each school with the head teachers' assistance. The children in the selected classes completed a two-page questionnaire, which took approximately 15 min. The questionnaires were filled out independently and they provided their personal information: gender, age, and type of residence (urban or rural). Children who completed the survey were rewarded with a bird watching festival sticker.

## Statistical analyses

Content analysis was used to identify and determine the themes of the focus group interviews. After analyzing the transcripts line by line, codes were generated from words or phrases. The derived categories were constantly compared with the codes that emerged from the subsequent set of data until no new themes were found (*Creswell, 2014*). Finally, we obtained three themes (direct, indirect, and vicarious nature experiences) and their categories and frequency were included (Fig. S1).

Questionnaire data were analyzed using IBM Statistics SPSS 22 and R 4.0.2 (R Foundation for Statistical Computing, www.rproject.org). Questionnaires with a completion rate of less than 50% were excluded, as were those with a large number of similar answers. Principal axis factor analysis with promax rotation largely confirmed the scales' theoretical constructs, and each item's factor loading was examined (Table 2 and Table S2). Construct validity, which determines whether the instrument truly measures the proposed hypothetical constructs, was established using the Kaiser-Myer-Olkin (KMO) test (*Cohen, Manion & Morrison, 2002*). Reliability among the scale items and subscales was measured using Cronbach's α, and the three subscales were acceptable (Table 2 and Table S2).

**Table 2 The reliability and validity of questionnaires (N = 2,175).**

| Variables | Items | Cronbach alpha | KMO | Mean ± SD | PCA (%) |
|---|---|---|---|---|---|
| Direct nature experience | 9 | 0.686 | 0.784 | 3.34 ± 0.85 | 49.91 |
| Indirect nature experience | 4 | 0.894 | 0.829 | 3.24 ± 1.17 | 70.17 |
| Vicarious nature experience | 4 | 0.763 | 0.765 | 3.45 ± 0.99 | 58.71 |
| Emotional connection | 7 | 0.754 | 0.847 | 4.25 ± 0.62 | 41.60 |
| Cognitive connection | 6 | 0.705 | 0.794 | 4.52 ± 0.51 | 41.33 |
| Pro-environmental behavior | 6 | 0.786 | 0.850 | 3.82 ± 0.73 | 48.72 |
| Pro-nature behavior | 7 | 0.840 | 0.842 | 3.03 ± 0.98 | 51.16 |

Variance analysis was used to further test the effect of individual characteristics on response variables. T-test and analysis of variance (ANOVA) were used for single variable variance analysis, and Tukey honestly significant difference (HSD) multiple comparisons correction with a 95% family wise confidence level was employed to compare the differences between age and location. The interaction effect conducted with the "HH" package in R was also fitted for each of the demographic variables to test whether there are condition-specific differences in the response variables.

A mixed linear model was used for analysis since the dependent variable was the average score, in which pro-environmental and pro-nature behaviors were separately used as the response variables. Individual characteristics (*i.e.*, gender, age, location, and residence), three categories of nature experiences (*i.e.*, direct, indirect, and vicarious), and connection with nature were used as explanatory variables while controlling school as a random factor. First, the variance inflation factors (VIFs) between the explaining variables were checked with the "car" package, and we found no evidence for multicollinearity (all VIFs < 2; *Gareth et al., 2013*). Seven regression models were conducted: Model 1 included all socio-demographic variables; Models 2, 3, and 4 separately involved the three types of nature experiences; Model 5 incorporated all experience types and considered their interactions; Model 6 included cognitive and emotional connections with nature; and Model 7 contained all socio-demographic and explanatory variables (Tables 3 and 4). We used the "lm4" package to perform a mixed linear model, called the "lmerTest," and the "MuMIn" package to obtain variable significance and the interpretability ($R^2$) of the overall model. We then performed model fitting and plotted the results using the "ggplot2" and "forestplot" packages in R 4.0.2 (*R Core Team, 2020*).

## Ethics

The research protocol of this study was reviewed and approved by the Biomedical Ethics Expert Committee of Xishuangbanna Tropical Botanical Garden at the Chinese Academy of Sciences (approval ID: XTBG-2020-9). Prior this research, we obtained informed consent from the schools and participants' parents. Verbal consent was also obtained directly from the children involved. The research purpose was explained to participants, and they were informed of their right to freedom of expression and to withdraw at any time. We did not judge the participants during the research process.

**Table 3 Model explaining children's *pro-environmental behavior* with nature experiences, connection with nature, and demographic variables.**

| | Model 1 | | Model 2 | | Model 3 | | Model 4 | | Model 5 | | Model 6 | | Model 7 | |
|---|---|---|---|---|---|---|---|---|---|---|---|---|---|---|
| | B (SE) | t | B (SE) | t | B (SE) | t | B (SE) | t | B (SE) | t | B (SE) | t | B (SE) | t |
| **Demographic variables** | | | | | | | | | | | | | | |
| Age-nine | 0.04 (0.05) | 0.90 | | | | | | | | | | | 0.01 (0.03) | 0.43 |
| Age-eleven | 0.03 (0.04) | 0.73 | | | | | | | | | | | 0.02 (0.03) | 0.64 |
| Age-twelve | −0.10 (0.04) | −2.36* | | | | | | | | | | | −0.05 (0.03) | −1.77 |
| Gender-girl | 0.08 (0.03) | 2.77** | | | | | | | | | | | −0.04 (0.02) | −1.92 |
| Residence-rural | −0.16 (0.13) | −1.26 | | | | | | | | | | | 0.08 (0.08) | 1.09 |
| Location-Hangzhou | 0.04 (0.15) | 0.28 | | | | | | | | | | | 0.02 (0.09) | 0.25 |
| Location-Kunming | 0.06 (0.15) | 0.40 | | | | | | | | | | | 0.04 (0.09) | 0.48 |
| **Nature experiences** | | | | | | | | | | | | | | |
| Direct | | | 0.46 (0.02) | 29.64*** | | | | | 0.25 (0.07) | 3.50*** | | | 0.13 (0.02) | 6.29*** |
| Indirect | | | | | 0.31 (0.01) | 25.32*** | | | 0.11 (0.06) | 1.87. | | | 0.08 (0.01) | 6.06*** |
| Vicarious | | | | | | | 0.43 (0.01) | 34.08*** | 0.38 (0.06) | 6.29*** | | | 0.23 (0.02) | 14.79*** |
| DE: IDE | | | | | | | | | 0.01 (0.01) | 0.24 | | | | |
| DE: VE | | | | | | | | | −0.02 (0.02) | −1.12 | | | | |
| IDE: VE | | | | | | | | | −0.01 (0.01) | −0.70 | | | | |
| **Connection with nature** | | | | | | | | | | | | | | |
| Cognitive | | | | | | | | | | | 0.36 (0.03) | 11.03*** | 0.25 (0.03) | 8.83*** |
| Emotional | | | | | | | | | | | 0.36 (0.03) | 13.16*** | 0.12 (0.03) | 4.82*** |
| R² (fixed variables) | 0.024 | | 0.290 | | 0.238 | | 0.349 | | 0.413 | | 0.263 | | 0.461 | |
| R² (total) | 0.121 | | 0.335 | | 0.307 | | 0.387 | | 0.449 | | 0.299 | | 0.495 | |

**Notes:**
* $p < 0.05$.
** $p < 0.01$.
*** $p < 0.001$.
Variable default Settings: Age (Default = 10), Location (Default = Xishuangbanna). B = standardized beta coefficients, SE = standard error.

# RESULTS

## Variation of different types of nature experiences

There was great variation in the frequency and type of nature experiences among the preadolescent individuals with regard to city *vs* rural, geographic location, and gender.
**Table 4 Model explaining children's *pro-nature behavior* with nature experiences, connection with nature, and demographic variables.**

| | Model 1 | | Model 2 | | Model 3 | | Model 4 | | Model 5 | | Model 6 | | Model 7 | |
|---|---|---|---|---|---|---|---|---|---|---|---|---|---|---|
| | B (SE) | t | B (SE) | t | B (SE) | t | B (SE) | t | B (SE) | t | B (SE) | t | B (SE) | t |
| **Demographic variables** | | | | | | | | | | | | | | |
| Age-nine | 0.11 (0.06) | 1.81 | | | | | | | | | | | 0.08 (0.05) | 1.74 |
| Age-eleven | 0.02 (0.06) | 0.39 | | | | | | | | | | | 0.03 (0.04) | 0.65 |
| Age-twelve | −0.13 (0.06) | −2.29* | | | | | | | | | | | −0.05 (0.04) | −1.14* |
| Gender-girl | 0.07 (0.04) | 1.82 | | | | | | | | | | | −0.12 (0.03) | −3.94*** |
| Residence-rural | −0.18 (0.12) | −1.46 | | | | | | | | | | | 0.16 (0.07) | 2.37* |
| Location-Hangzhou | −0.17 (0.15) | −1.15 | | | | | | | | | | | −0.23 (0.08) | −2.67 |
| Location-Kunming | −0.08 (0.15) | −0.52 | | | | | | | | | | | −0.11 (0.08) | 1.29 |
| **Nature experiences** | | | | | | | | | | | | | | |
| Direct | | | 0.69 (0.02) | 34.25*** | | | | | 0.47 (0.09) | 5.03*** | | | 0.28 (0.03) | 10.40*** |
| Indirect | | | | | 0.45 (0.02) | 28.04*** | | | 0.11 (0.08) | 1.37 | | | 0.13 (0.02) | 7.35*** |
| Vicarious | | | | | | | 0.59 (0.02) | 34.26*** | 0.27 (0.08) | 3.46*** | | | 0.27 (0.02) | 12.77*** |
| DE: IDE | | | | | | | | | −0.02 (0.02) | −1.05 | | | | |
| DE: VE | | | | | | | | | −0.01 (0.02) | −0.56 | | | | |
| IDE: VE | | | | | | | | | 0.03 (0.02) | 1.53 | | | | |
| **Connection with nature** | | | | | | | | | | | | | | |
| Cognitive | | | | | | | | | | | 0.28 (0.05) | 6.08*** | 0.12 (0.04) | 3.18** |
| Emotional | | | | | | | | | | | 0.62 (0.04) | 16.50*** | 0.26 (0.03) | 7.69*** |
| R² (fixed variables) | 0.022 | | 0.354 | | 0.283 | | 0.358 | | 0.455 | | 0.249 | | 0.491 | |
| R² (Total) | 0.070 | | 0.393 | | 0.326 | | 0.372 | | 0.485 | | 0.269 | | 0.504 | |

Notes:
* $p < 0.05$.
** $p < 0.01$.
*** $p < 0.001$.
Variable default Settings: Age (Default = 10), Location (Default = Xishuangbanna). B = standardized beta coefficients, SE = standard error.

Urbanization significantly influenced preadolescents' nature experiences. For some direct experiences, the scores were relatively low, such as, catch fish and tadpoles = 2.29 ± 1.26; climb trees = 1.90 ± 1.23; pick fruits = 2.20 ± 1.30; play mud = 2.11 ± 1.28 (Fig. 1, Table 2), with respondents currently residing in urban areas having significantly lower scores than those in rural areas. However, children in urban settings reported some more frequent direct experiences (planting flowers and trees, mountaineer, fly a kite, collect natural things, and observe insects) than rural children. Urban children clearly had significantly

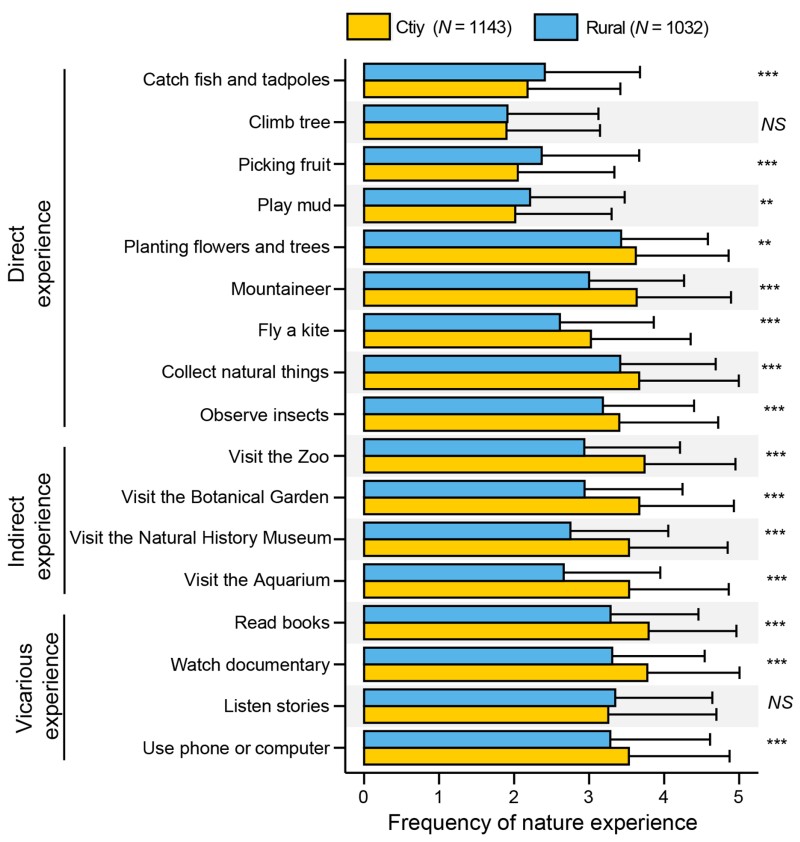

**Figure 1 Composition of preadolescents' nature experiences: city vs rural.** The bar chart reflects mean values and standard deviations for each category. Frequency of nature experiences: Never = 1, Seldom = 2, Sometimes = 3; Often = 4; Always = 5. Significance levels: ***$p < 0.001$; **$p < 0.01$.

higher indirect and vicarious experiences than rural children, except for one aspect (listen to natural stories) (Fig. 1). We also found significant, positive, and high correlations between different types of nature experiences (Table S5; 0.5< Pearson correlation coefficient <0.7).

Location plays an important role in influencing children's natural experiences. Children from the Kunming areas reported higher scores for direct and indirect nature experiences than those in Xishuangbanna. However, there was no significant difference in the scores of vicarious nature experiences among the three areas (Fig. 2A). The interaction between location and place of residency was significant ($F$ (direct experience) = 10.76, $p < 0.001$; $F$ (indirect experience) = 8.058, $p < 0.001$; $F$ (vicarious experience) = 19.921, $p < 0.001$; Fig. 2B). Urban preadolescents in Kunming and Xishuangbanna reported higher scores for nature experiences (direct, indirect, and vicarious) than rural children, but this was not the case for Hangzhou, where no significant difference was found in direct nature experiences between rural and urban areas.

After taking into account other socio-demographics, we found that children's natural experiences differed by gender and age (Tables S3 and S4). Girls' self-reported direct, indirect, and vicarious nature experience frequencies were significantly higher than those

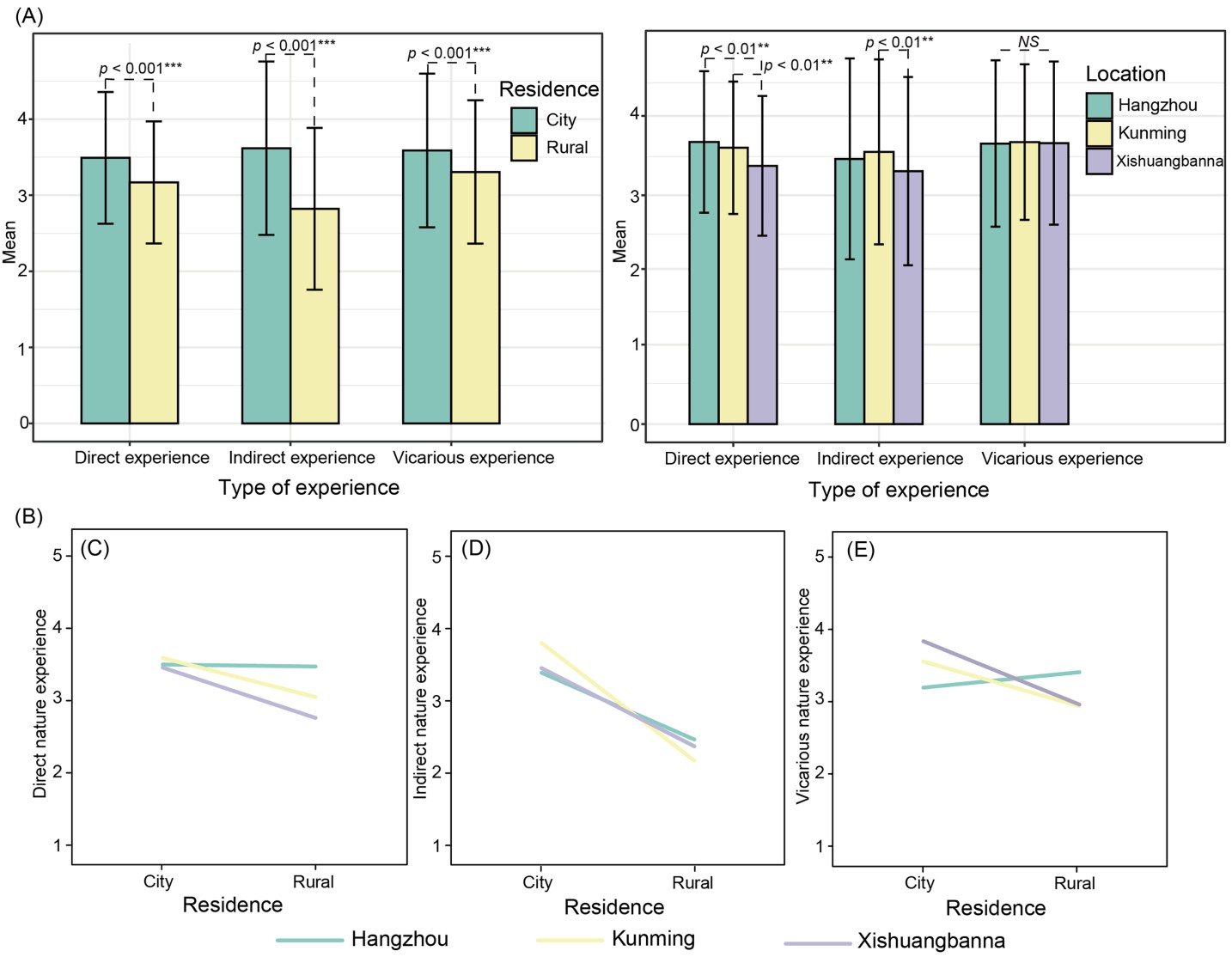

**Figure 2 T-test (A) and ANOVA (B) indicate the effect of geographic variations (location and residence) on nature experiences.** The error bars on the bar chart represent the standard deviation. The interaction plots show predictors for (C) direct nature experience, (D) indirect nature experience, and (E) vicarious nature experience.

of boys were. Direct and indirect nature experiences tended to decline with age, although no significant change was found in vicarious experiences. Children aged 9 and 10 years had a higher frequency of direct experience than 11 and 12 year-olds, while 12-year-old children's indirect natural experience was significantly lower than that of children of other ages.

## Variation of connection with nature and conservation behavior

Similar to nature experiences, rural preadolescents reported lower scores on connection with nature and conservation behavior than urban children. Meanwhile, there was a significant difference among locations in cognitive connection: Xishuangbanna preadolescents had lower scores than those in Hangzhou and Kunming. However,

pro-nature behavior was significantly lower in Hangzhou compared to children in Kunming and Xishuangbanna (Fig. 3A). The interaction between the type of location and place of residency was significant ($F$ (cognitive connection) = 25.573, $p < 0.001$; $F$ (emotional connection) = 15.191, $p < 0.001$; $F$ (pro-environmental behavior) = 50.257, $p < 0.001$; $F$ (pro-nature behavior) = 16.479, $p < 0.001$; Fig. 3B). Respondents living in the cities of Kunming and Xishuangbanna scored higher on connection with nature and conservation behaviors than rural preadolescents, but the opposite was true in Hangzhou.

Girls reported significantly higher scores than boys in both emotional and cognitive connections with nature and self-reported pro-environmental behavior (Table S3). Moreover, while we did not find any significant difference in connection with nature across ages, 12-year-olds had significantly lower conservation behavior than the other age groups (Table S4).

### The relationship between nature experiences, connection with nature, and conservation behavior

The results of the mixed linear model showed that both conservation behaviors were mainly explained by preadolescents' nature experiences and connection with nature (Tables 3 and 4). Direct, indirect, and vicarious nature experiences alone were all significantly correlated with pro-environmental and pro-nature behaviors (Tables 3 and 4, Fig. 4). Collectively, all three types of nature experience explained 44.9% and 48.5% of pro-environmental and pro-nature behaviors, respectively. More specifically, direct and vicarious experiences had stronger effects on pro-nature behavior than indirect experiences, and vicarious nature experiences exhibited a stronger effect on pro-environmental behavior than direct and indirect experiences (Fig. 4). Meanwhile, connection with nature was significantly positive correlated with conservation behavior (Tables 3 and 4, Fig. 4).

## DISCUSSION

This study showed that the types and contents of nature experiences of contemporary Chinese preadolescents varied greatly, and both location and the type of residency (urban *vs* rural areas) significantly affected children's nature experiences. All three types of nature experiences (direct, indirect, and vicarious nature) alone are positively correlated with conservation behavior and collectively predict approximately 45% of self-reported conservation behavior. Thus, different kinds of nature experiences are deemed important in shaping modern children's conservation behaviors in China.

### Undergoing change of nature experiences

For many preadolescents, nature experiences are often a transformation of one kind of interaction (direct) to another (indirect or vicarious) (*Clayton et al., 2017*). Our study showed that contemporary children's experience of nature are diverse and varied; some direct experiences were less frequently observed among urban respondents, while they had a higher frequency of indirect and vicarious experiences compared to children living in rural areas. Previous studies have shown a cross-generational decline in the experience of

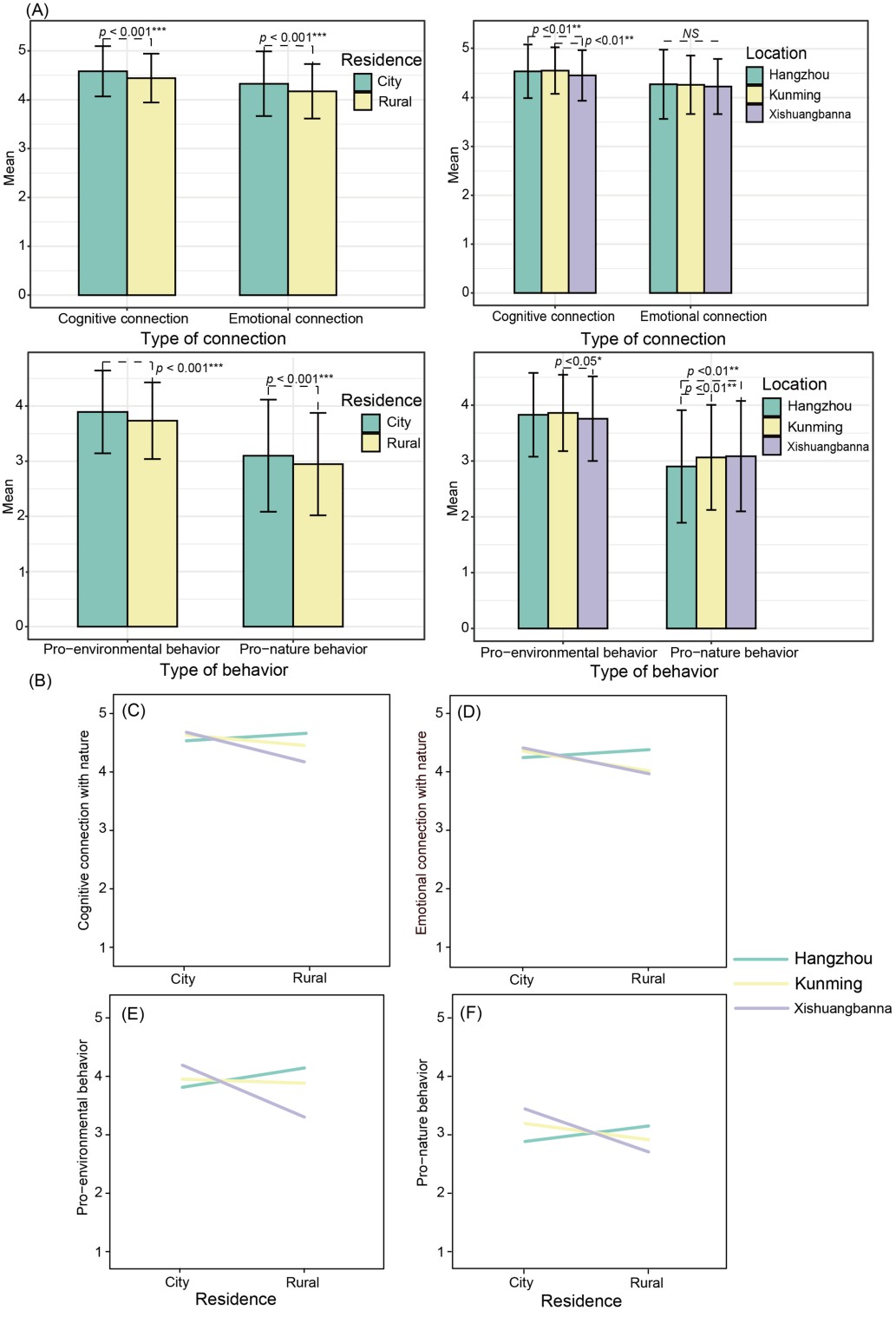

**Figure 3 T-test (A) and ANOVA (B) indicate the effect of geographic variations (location and residence) on predictors.** The error bars on the bar chart represent the standard deviation. The interaction plots show predictors for (C) cognitive connection with nature, (D) emotional connection with nature, (E) pro-environmental behavior, and (F) pro-nature behavior.

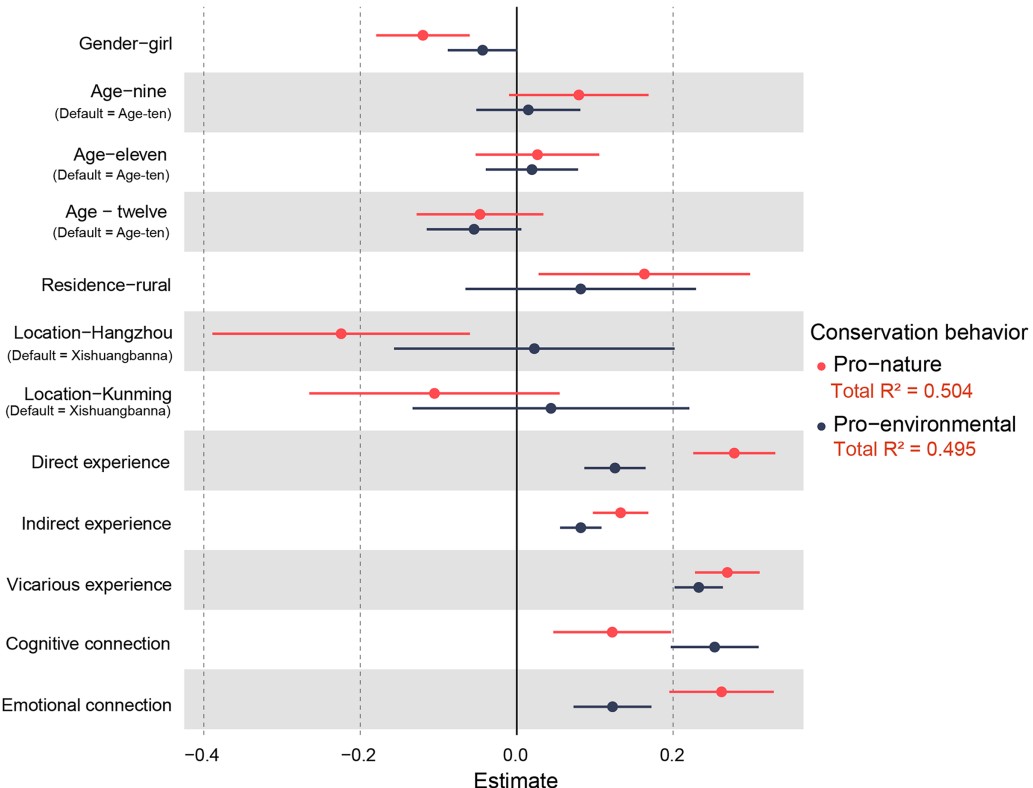

**Figure 4** Regression coefficients for all explanatory variables correlated with conservation behavior. For each estimate, circles denote the averaged beta coefficient and horizontal line show the 95% CI. When error bars cross the zero-estimate line, the estimates are considered not significant.

neighborhood flowering plants (*Soga & Gaston, 2018*; *Novotný et al., 2021*) and extensive farming or taking care of field animals (*Novotný et al., 2021*). However, there is also evidence suggesting an increasing trend for direct interactions with some flora and fauna groups (*Imai, Nakashizuka & Kohsaka, 2019*), gardens, and forests across time. Shifts in nature experiences may be linked to changes in children's daily routines, in which there is a limited radius for autonomous activities, especially for children living in inner-city areas (*Kyttä et al., 2015*; *Martens & Molitor, 2020*). Additionally, our study found that children's vicarious experiences did not differ significantly across locations (Fig. 2A). One possible reason for this result could be the broader trends of spending more time on sedentary behaviors (*Ding et al., 2020*) and extended screen time among adolescents (*Edwards & Larson, 2020*). Another possible interpretation relies on ongoing culturally driven transformations (*Novotný et al., 2021*) that strongly reflect the equality in formal education (*Zhang, Qin & Liu, 2019*) or the widespread access to books and media across China.

In addition to the variation in different types of nature experiences, we also found significant interaction effects between location and residency. Although urban preadolescents living in Kunming and Xishuangbanna reported higher scores for nature experiences (direct, indirect, and vicarious) than rural children, the same pattern was not present in the Hangzhou area. Previous studies have reported that children raised in rural

settings do not necessarily have greater access to natural resources than urban dwellers (*Hinds & Sparks, 2008*). A few studies explain how rural children's nature experiences may differ from those of urban children, especially when the "countryside" becomes part of their lifestyle and identity (*King & Church, 2013*). It is possible that children from higher socioeconomic status will have better access to green spaces (*Wolch, Byrne & Newell, 2014*), which could contribute to a dual environmental identity among well-to-do city children. This identity is characterized by an attraction to distant environments, such as the countryside, which are often idealized as more natural and desirable than the urban environments in which they live (*Kyttä, 2002*).

## Nature experiences promote conservation behaviors

This study indicated that all three types of nature experiences showed significant predictability for self-reported conservation behaviors. Direct experiences significantly predicted children's conservation behaviors, particularly pro-nature behavior (Fig. 4). This finding supports the widely held viewpoint that childhood experiences include bodily interaction and sensory attentiveness and may potentially stimulate curiosity, discovery, eco-awareness, and therefore biodiversity conservation willingness (*Linzmayer & Halpenny, 2013*; *Beery & Jørgensen, 2018*; *Beery & Lekies, 2019*; *Richardson et al., 2020*; *Abdullah, Ishak & Ahmad, 2022*). Moreover, direct experience of nature, especially that with explicit attentiveness to biodiversity, shifted participants' views in relation to nature, which acted as a catalyst for participants to engage in pro-biodiversity behavior (*Cosquer, Raymond & Prevot-Julliard, 2012*; *Prévot et al., 2018*). Thus, direct nature experiences may make biodiversity experientially accessible to people and raise personal relevance to nature.

Although the effect size of indirect nature experiences on conservation behavior was relatively lower than that of direct nature experiences, indirect experiences still significantly predicted conservation behavior. Earlier studies indicated that participants with interactive experiences in botanical gardens or zoos gained more knowledge of biodiversity, were more connected to nature, and even improved their interaction behavior compared to general visitors (*Jensen, 2014*; *Moss, Jensen & Gusset, 2015*; *Zelenika et al., 2018*; *Collins et al., 2020*). On the other hand, botanical gardens and zoological facilities are considered to be human-designed places that may curate mini-representations of nature (*Heyd, 2006*). Visitors, including children, prefer to see animals in naturalistic environments instead of traditional zoo enclosures (*Moss & Pavitt, 2019*). Nevertheless, our study provides evidence for the positive effects of indirect nature experiences on preadolescents' conservation behaviors in China.

Notably, vicarious nature experiences significantly predicted both pro-environmental and pro-nature behaviors. This finding is in line with other recent studies suggesting that vicarious nature experiences can be the primary information sources for some individuals to learn about the environment (*Boubonari, Markos & Kevrekidis, 2013*; *Gaston & Soga, 2020*) and that mediated experience widely complements and expands real experience (*Fletcher, 2017*; *Klofutar, Jerman & Torkar, 2020*). Such more passive activities of looking at nature-related books or websites and making nature the topic of conversation with others reflect greater personal and purposeful involvement, which may have a synergistic

effect on boosting people's pro-nature conservation behavior (*Richardson et al., 2020*). Perhaps this phenomenon can be explained by the changing narrative style in vicarious experiences. Although affective storytelling and arresting aesthetics remain central (*Somerville et al., 2021*), there is an apparent trend toward more explicit conversations around biodiversity loss and anthropogenic causes of ecological change (*Aitchison, Aitchison & Devas, 2021*). Effective emotional engagement and storytelling, as well as the degree of immersion or technological novelty, may therefore be key to the immediate impact of vicarious nature experiences (*Blythe et al., 2021*).

Interestingly, our research also found high correlations among different nature experiences, which suggests that the more direct nature experiences children have, the more indirect and vicarious experiences they engage in, and vice versa. Combinations of types of nature experiences can generate a collective effect on conservation behaviors. Previous studies have indicated that indirect and vicarious experiences widely complement and expand real experiences (*Almeida, Fernández & Strecht-Ribeiro, 2018*; *Fletcher, 2017*) and can even partially satiate the need for contact with nature (*Kahn, Severson & Ruckert, 2009*). Another study also demonstrated that elementary school children's direct and vicarious nature experiences positively influenced their affective attitudes and willingness to conserve biodiversity (*Soga et al., 2016*). Therefore, it is plausible that various forms of experiences increase the number and diversity of opportunities for exploring in natural environments (*Klofutar, Jerman & Torkar, 2020*), which, in turn, stimulates multiple sensory channels for learning and presents information in a more salient way than learning *via* a single sensory channel (*Li et al., 2022a*).

## Connection with nature positively influence conservation behaviors

Diverse nature experiences have specific consequences for conservation behaviors, and these associations might be mediated *via* connections with nature. Our results support that connection with nature positively influences conservation behavior. Consistent with earlier studies, connection with nature and engagement in nature activities occupied the largest portion of the explained variance in pro-nature behavior (*Richardson et al., 2020*). Additionally, general connection with nature was strongly associated with engagement in pro-environmental activities such as recycling and turning off lights (*Clayton & Karazsia, 2020*). Previous studies also found that changes in pro-conservation behavior after visiting a zoo may be related to personal connections to animals (*Skibins, Powell & Hallo, 2013*), and nature documentary exposure increased pro-environmental donation behavior only in those already possessing a strong sense of connectedness (*Arendt & Matthes, 2016*). One possible interpretation could be that psychological and physical connections with nature represent deeply personal relevance that can drive an inherent motivation to act, sustain, and foster conservation (*Knapp & Benton, 2006*; *Mayer et al., 2009*).

## Other factors affecting conservation behaviors

Our study revealed that differences in nature connectedness and conservation behavior vary according to the intertwined degree of urbanization and biodiversity. Urban preadolescents from Kunming and Xishuangbanna scored higher on connection with
nature and conservation behaviors than their rural counterparts, but the opposite was true in Hangzhou (Fig. 3B). Similar findings have been reported in previous studies that found no significant differences in species identification knowledge among children despite different levels of urbanization (*Schlegel, Breuer & Rupf, 2015*; *Bashan, Colléony & Shwartz, 2021*), and that 4th–8th grade children's attitudes toward animals are similar across suburban, exurban, and rural areas (*Schuttler et al., 2019*). It is worth noting that while the qualities of variation and diversity in biotic/abiotic features of spatial proximity access are important, this does not imply that only vast and wild settings can meet these access needs (*Giusti, Barthel & Marcus, 2014*). Along with urbanization, a series of new habitats in urban areas are being created, such as road verges, parks, and gardens (*Li et al., 2019*). Children may engage in ample nature experiences within the limitations afforded by cities (*Almeida, Fernández & Strecht-Ribeiro, 2018*; *Freeman et al., 2018*), in turn drawing attention to natural components and knowledge (*Sampaio et al., 2018*). Moreover, no significant effects of biodiversity gradients were found, which might be because the items listed to measure nature experiences rarely link to biodiversity. Alternatively, perhaps the wildlife in surrounding areas is often overlooked in conservation education (*Hanisch, Johnston & Longnecker, 2019*).

### Limitations and prospects for future research

Our study distinguished between three types of nature experiences of modern Chinese preadolescents and further explored the unique contributions of different experiences in supporting conservation behaviors. Given that global efforts to protect nature need to consider cultural diversity, we provide nuanced evidence from Chinese children. As nature experience and interpretation can be influenced by cultural factors, it is important to exercise caution when generalizing our conclusions beyond China. Additionally, certain limitations of our study should be taken into consideration when interpreting the results. While focus group interviews can create a comfortable environment for verbal dialogue with children, it is crucial to acknowledge that children's opinions may be influenced by their peers, and some children who are less vocal may feel inhibited or silenced.

Our methodology mainly relied on self-reports, which can always be biased due to social desirability. Although we carried out a large-scale survey, the overall study sample was not sufficiently representative, and other possible nature experiences and behavior in other parts of China were not taken into account. In addition, the cross-sectional data of nature experiences cannot assess the extent of changes; instead, they speculatively obtain the result of the change. We are also unable to ascertain the existence or direction of causality between nature experiences, connection with nature, and conservation behavior owing to limitations in our research methods. The role of parents or other significant individuals in shaping children's conservation behaviors was also not considered in this study, therefore, these issues should be further investigated in future research.

### CONCLUSIONS

We specifically examined the causal relationship between nature experiences, connection with nature, and concerning conservation behavior. Our research recognizes the positive

role of direct, indirect, and vicarious experiences in promoting preadolescents' conservation behaviors. The essential path seems to create interactive and meaningful experiences with deep personal relevance. Promoting meaningful and integrative nature experiences, building connections, and taking action to preserve nature could be effective strategies for children's environmental education (*Thomas et al., 2019*).

## ACKNOWLEDGEMENTS
We are very grateful to the schoolchildren and their school principals for participating in the survey.

### Funding
The authors received no funding for this work.

### Competing Interests
The authors declare that they have no competing interests.

### Author Contributions
- Zhihui Yue conceived and designed the experiments, performed the experiments, analyzed the data, prepared figures and/or tables, authored or reviewed drafts of the article, and approved the final draft.
- Jin Chen conceived and designed the experiments, analyzed the data, authored or reviewed drafts of the article, and approved the final draft.

### Human Ethics
The following information was supplied relating to ethical approvals (*i.e.*, approving body and any reference numbers):

The research protocol of this study was reviewed and approved by the Biomedical Ethics Expert Committee of Xishuangbanna Tropical Botanical Garden at the Chinese Academy of Sciences (approval ID: XTBG-2020-9).

### Data Availability
The raw data and code are available in the Supplemental Files.

### Supplemental Information
Supplemental information for this article can be found online at http://dx.doi.org/10.7717/peerj.15542#supplemental-information.

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
