# Peer review of "Direct, indirect, and vicarious nature experiences collectively predict preadolescents’ self-reported nature connectedness and conservation behaviors"

_PeerJ, doi:10.7717/peerj.15542_

## Round 0.1 · original submission · Major Revisions

· Academic Editor

Major Revisions

Dear Dr. Yue,

Thank you for submitting your manuscript to PeerJ. Your paper was evaluated by three individuals with considerable expertise in this area. All three reviewers have found your research question and study very interesting and have commented on a number of significant strengths. At the same time, they have identified several concerns. Based on my independent reading, I agree that your paper has potential for an impactful contribution, but major revisions are necessary. Thus, I am inviting you the option of revising and resubmitting it for consideration.

The reviewers have provided exceptionally thorough and constructive comments. I will not repeat their points but I urge you to attend to all of their points, especially those related to the validity of the findings and your statistical analyses. You are welcome to submit a revision, if you think that you can address all concerns.

Thank you for giving us the opportunity to consider your work, and we wish you all the best with your article.

Yours sincerely,

Andree Hartanto, PhD
Academic Editor
PeerJ

·

Basic reporting

In this manuscript, the authors explore the relationship between three different forms of nature experiences, direct (e.g., climbing trees, catching animals), indirect (e.g., visiting zoos, aquariums) and vicarious (e.g., reading books, listening to stories etc), and nature conservation behaviors. This relationship was explored specifically in pre-adolescent children from 3 regions of China, where each region consisted of a mix of urban and rural populations.

The paper is well-written in clear English, and a good amount of background information is provided. I support the acceptance of this manuscript, albeit with some revisions.

Major comments:
1. Figure S1 is an important figure that lists the different kinds of behaviors with size of the text the same as the size of the text. However, the word cloud is a rather uninformative way of putting it. I suggest reporting the behaviors in tabular form with some form of frequency metric to describe the raw occurrence numbers for each experience

2. Line 357: Regarding the sentence, "For some direct experiences, the scores were lower than the average frequency". Here, I understand that you mean total score for urban and rural population combined. It would be good to clarify that

3. In general, there is little information about how one of the central variables in this manuscript, the "pro-nature behavior" has little quantification information provided. It is first mentioned in Line 350. How were these behaviors defined? And how were they quantified? Similar quantification question about conservation behaviors.

There are some minor comments such as grammar errors etc that are worth fixing:

1. Lines 133: ...key to "gaining" instead of "gain". Although both "gain" and "gaining" are grammatically correct, the larger goal is clearly answered with gaining

2. Line 432: "some direct experiences were seldom observed among urban respondents". This is in relation to Figure 1. However, Figure 1 shows that each direct experience was observed, at least once, in both urban and rural respondents. No direct experience had zero occurrence. So "seldom" is not the appropriate term here. I would reword to "less frequently"

3. Line 475: ...improved their "interact" behavior -> improved their "interaction" behavior

4. Line 335: The reference is inaccurate. There is no "Figure S2". I believe the authors meant "Table S2"

Experimental design

In general, the analyses done in this manuscript are reasonable. The authors make adjustments to critical values when making multiple measurements to account for the possibility of making more false positive judgements.

The research question is also well laid out. However, there are some places in the text which should be clarified further. Specifically, around lines 260, the authors write that "Thirteen focus group discussions (N=156)" were held. Since each focus group had 12 participants (mentioned in Line 254), and there were 4 schools, it appears as if there were several focus groups within each school. Please provide a breakdown of how many focus groups were present in each school.

Validity of the findings

The findings are valid and make sense to me. However, there is a limitation of this study that arises from interviewing students in focus groups instead of one-on-one discussions. It is reasonable to posit that children's answers would be dependent on peers, e.g., due to peer pressure to assimilate or derive answers from friends etc. Due to that reason, it is possible that the results would not hold truly if the survey was conducted 1:1.

I am not suggesting re-conducting the survey, rather I believe that this point should be brought up in the Discussion section and in the Limitations section as being a possible factor

Reviewer 2 ·

Basic reporting

Overall, the writing is easy to understand and the literature review covers sufficient background.

• Line 26: “Thus, further study of potential impacts on future conservation willingness is needed.” Impacts of what? I understand what the authors mean, but this statement should be clearer.

• Line 35: “Emotional and cognitive connection with nature positively predicted conservation behavior, mediated by location and residence type”. I’m not sure if the term mediation is used correctly here. The statement implies that location lies on the causal path between nature connectedness and behavior. This doesn’t make sense theoretically and as not reflected in the analysis either.

• I suggest moving the introduction of connection to nature (line 170 onwards) to earlier in the paper as the construct was mentioned in the opening paragraphs but not fully explained. By understanding the construct first, readers can form their own opinions and speculations of how different nature experiences may promote connection to nature.

• Line 62: “Diminished human experience of nature, specifically of biodiversity”. Are there nature experiences which do not involve experiences of biodiversity? Do the authors mean experiences of wild nature vs. curated/indoor nature?

• Line 107: Authors may want to highlight that thanks to simulated nature experiences, there may not necessarily be “extinction of experience” (Soga and Gaston, 2016). It is just that the type and context of experiences may have shifted.

Experimental design

It is nice to see a sample from both urban and rural areas in China. The study design is sound with a few points for clarification. However, the data analysis does not seem to cover as much ground as the introduction alludes to, particularly with regard to predicting nature connectedness.

• Analysis: I’d be interested in seeing a model that uses nature experiences to explain variances in connection with nature. Such an analysis was implied by the paper title “… nature experiences collectively predict preadolescents’ self-reported nature connectedness…” as well as study purpose. As the authors have acknowledged, connection with nature often mediates the relationship between nature experiences and environmental behavior (e.g., Cheng & Monroe, 2010). Do all types of nature experience promote nature connection?

• Ideally, there should be a mediation model with indirect effects (experience  connection  behavior), but I understand this may not be common practice in fields outside of psychology.

• Line 226: “We opportunistically sampled four primary schools in Xishuangbanna”. Do these 4 schools cover both urban and rural contexts? This is important for ensuring that the nature experiences reported are exhaustive.

• What criteria did the authors use to determine which samples are from urban vs. rural areas? Sometimes, small townships are considered suburban or rural. So it’s good to clarify the criteria.

Validity of the findings

The discussion needs a paragraph on how the findings may or may not be generalizable to populations outside China. My opinion is that there are no major issues with generalizability. However, the authors may want to clarify if China has any environmental education programs (e.g., field trips, ecology classes) that are implemented widely in schools. If there is such a thing, it would mean that urban children are exposed to nature not because their lifestyles encourage it, but because it was enforced.

Additional comments

Thank you for the opportunity to review this article. I think the paper has potential to contribute to literature on nature experiences (which are indeed very diverse), provided some issues of cohesiveness between the introduction and analysis are addressed.

Reviewer 3 ·

Basic reporting

The current manuscript examines the relationship between direct, indirect, and vicarious nature experiences on pre-adolescents’ self-reported nature connectedness and conservation behaviours. In my view, this manuscript deals with a very interesting and important issue of environmental conservation behaviours amongst children, which is a very crucial sample, and I believe it meaningfully extends upon the existing literature. That being said, there are several limitations that the authors should consider addressing.


Abstract/Introduction

1. Under the results section of the abstract (line 36), you mention that emotional and cognitive connection with nature positively predicted conservation behaviour, mediated by location and residence. However, there was no mediation analyses reported in the study. Authors should be careful in their phrasing.

2. The introduction could have benefitted from referencing prior research that informed the development of this study. While this is an interesting study, it is important to provide the rationales for why you are examining this specific research question.

3. The flow of the introduction would be improved by mentioning why you are specifically targeting children as your sample much earlier on in your introduction, rather than referencing a very broad term “younger generations” (line 61). What are the implications of investigating this study using a sample of pre-adolescent children, compared to say teenagers or even young adults? This could be discussed in greater detail.

a. Furthermore, you could also bring up that there is a huge disconnect between modern-day children and nature (see Bragg et al., 2013 and Sobel, 2017), which is extremely worrying from a natural and environmental standpoint.

4. In line 184: “…before the age of 12 generally have sustained effects later in life” – this sentence could do with greater clarification. Some examples that you could refer to for elaboration:

a. Children with positive experiences in nature have a deeper relationship with nature in adulthood (e.g. Charles et al., 2018 and Rosa et al., 2018)

b. Positive engagement with nature in childhood may predict greater likelihood in engaging in pro-environmental behaviour in adulthood (e.g. Evans et al., 2018 and Molinario et al., 2020)

5. I appreciated the immense detail and depth that you went into elaborating on what direct, indirect, and vicarious nature experiences are. However, I believe it would be helpful if there was some theoretical ground or scholarly speculation as to why you believe those three types of nature experiences might predict nature connectedness and environmental behaviours.

6. What are the hypotheses for this study? If there are no hypotheses to be made, this paper should be framed as an exploratory study.

7. This manuscript would generally benefit from additional proof reading for major grammatical errors and awkward sentence construction (e.g., see lines 62-64, line 268).

Experimental design

1. The focus group interviews were only carried out in schools located in Xishuangbanna. Why was this so (instead of extending it to schools in Kunming and Hangzhou)? I think it would be helpful to clarify the rationale for this.

2. In Table S1, I noticed that there was no report of biodiversity or urbanization density, yet you mentioned in lines 228-229 that the gradients of urbanization and biodiversity were considered. It would be informative if you could provide information on the biodiversity and urbanization densities of the cities listed in Table S1 (if such information is available). Otherwise, please consider rephrasing lines 228-229.

3. Two pilot tests were conducted; however, the results of these pilot tests were not reported. Additionally, was test-retest conducted when testing the reliability of the scales? This is especially important for newly created scales. How were the items and expressions readjusted? It would be good if you could report the results prior to the adjustment and after adjustment, as well as the original items and wording of expressions.

4. What was the purpose of extracting children’s frequently mentioned nature experience items from focus group interviews in the development of the nature experience scale? This is not quite clear. What was the rationale for combining two different scales to develop the nature experience scale? That is also not quite clear.

5. 17 items were kept from the initial nature experience scale that contained 20 items (line 276). It would be informative to report the 3 items that were dropped, as well as the statistical information that supports the dropping of the 3 items.

6. Cronbach’s alpha should be reported for each scale used.

7. The N of the sample used for the main experiment should be reported in the Methods section, rather than in the tables located in the appendix.

Validity of the findings

1. Items factor loading was reported. However, I note that there was no testing of various model fits (e.g., does a 2-factor model fit better than a 1-factor model). Model fit comparison should be reported (chi-square), together with fit indices for each model, which include AIC, BIC, RMSEA, SRMR, TLI and CFI (for an example of model fit comparison, see Toh and Yang, 2022, page 653). These are important when investigating factor structure.

Additional comments

1. You mentioned that the study showed how contemporary children’s experience of nature has changed significantly (lines 430-431). Be careful with the phrasing – the study wasn’t conducted longitudinally, and based on the results alone, we cannot say for sure that children’s experience of nature has changed significantly.

2. The results found that children in urban areas reported higher scores for nature experiences. This could be because people of higher SES have greater access to green spaces (e.g., Jennings et al., 2017; Rigolon et al., 2018; Wolch et al., 2014). It would be helpful if there was greater discussion on this.

3. Another important factor that affects conservation behaviour is parental attitude and behaviour (e.g., Barrable & Booth, 2020; Evans et al., 2018; Giusti et al., 2018; Passmore et al., 2020) which was left out in the discussion.

---

## Round 0.2 · Minor Revisions

· Academic Editor

Minor Revisions

Dear Dr. Yue,

Thank you for submitting your manuscript to PeerJ. We have just obtained the reviews from our experts. I also have read the manuscript myself independently before looking at the reviews. Overall, we are satisfied with the revision and agree that your paper has potential for an impactful contribution.

One of the reviewers also raised a number of relatively minor concerns that should be easy for you to address. Pending the minor revision, I am happy to conditionally accept your paper for publication in PeerJ

Thank you for giving us the opportunity to consider your work, and we wish you all the best with your article.

Yours sincerely,

Andree Hartanto
Academic Editor
PeerJ

Reviewer 2 ·

Basic reporting

Line 119: Factors “that” influence conservation behaviors

Line 234: 3) residence type and geographic location may play moderating role in the relationship between nature experience, nature connectedness, and conservation behavior.”
• This statement is unclear. Grammar/phrasing also can be improved.
• Are there expected directions of the moderation effects? What is the theory behind such a speculation?
• In general, please explicitly link findings back to the three newly added hypotheses.

Line 383:” For some direct experiences, scores were lower than the average frequency for children's overall nature experience (catch fish and tadpoles = 2.29 ± 1.26; climb trees = 1.90 ± 1.23; pick fruits = 2.20 ± 1.30; play mud = 2.11 ± 1.28; Figure 1), with respondents currently residing in urban areas having significantly lower scores than those in rural areas
• Not sure what “average frequency for children’s overall nature experience” refers to.

Experimental design

No comment

Validity of the findings

Line 442: “More specifically, direct and vicarious experiences had stronger effects on pro-nature behavior, and vicarious nature experiences had a stronger effect on pro-environmental behavior (Figure 4).”
• I appreciated that the authors also discussed effect size and variance explained rather than just looking at significant results. However, based on figure 4, the confidence intervals of the beta coefficients (particularly indirect and vicarious experiences) overlap. So I’m not sure whether the claim that there is a “stronger effect” holds.

Additional comments

Overall, I found the revised manuscript to be substantially improved. The authors provided sufficient details to address the clarification questions. Compared to the previous version, the reporting and claims made in the revised manuscript are more congruent with the analyses conducted. I don’t find any remaining major concerns that may compromise on validity of the findings. With that said, I spotted a more points the authors may consider addressing.

Reviewer 3 ·

Basic reporting

I think the reviewer comments have been sufficiently addressed.

Experimental design

I think the reviewer comments have been sufficiently addressed.

Validity of the findings

I think the reviewer comments have been sufficiently addressed.

---

## Round 0.3 · accepted · Accept

· Academic Editor

Accept

Dear Dr. Yue,

I am pleased to advise that the above paper has now been accepted for publication in PeerJ. Thank you for giving the Journal the opportunity to publish your work. We are impressed with your paper and believe that it will contribute well to the literature. Well done!

Best Regards,
Andree